# Lorentz microscopy of optical fields

**John H. Gaida** [1,2], **Hugo Lourenço-Martins**[1,2], **Sergey V. Yalunin**[1,2], **Armin Feist** [1,2], **Murat Sivis** [1,2], **Thorsten Hohage** [3], **F. Javier García de Abajo** [4,5] & **Claus Ropers** [1,2] ✉

In electron microscopy, detailed insights into nanoscale optical properties of materials are gained by spontaneous inelastic scattering leading to electron-energy loss and cathodoluminescence. Stimulated scattering in the presence of external sample excitation allows for mode- and polarization-selective photon-induced near-field electron microscopy (PINEM). This process imprints a spatial phase profile inherited from the optical fields onto the wave function of the probing electrons. Here, we introduce Lorentz-PINEM for the full-field, non-invasive imaging of complex optical near fields at high spatial resolution. We use energy-filtered defocus phase-contrast imaging and iterative phase retrieval to reconstruct the phase distribution of interfering surface-bound modes on a plasmonic nanotip. Our approach is universally applicable to retrieve the spatially varying phase of nanoscale fields and topological modes.

The primary source of contrast in transmission electron microscopy (TEM) is elastic scattering at magnetic and electric potentials in an investigated specimen. Atomic-scale and elemental variations of the Coulomb potential imprint a phase profile onto a transmitted electron wave function. Transforming this Aharanov-Bohm phase[1] to measurable intensities, contrast is obtained by defocused imaging, beam deflection (differential phase contrast), off-axis holography or the use of Zernike phase plates[2–4]. In particular, Lorentz microscopy encompasses TEM variants designed to image phase shifts induced by nanoscale fields in electric and magnetic devices, primarily involving spin textures containing domain walls[5], vortices[6] and skyrmions[7,8].

These approaches were previously extended to image time-dependent potentials, employing pulsed electron beams synchronized to an external excitation[9–13]. At sufficiently low frequencies, typically in the radio-frequency range, the electron beam samples the instantaneous potential, with just a marginal net change in electron energy. In contrast, when an electron's interaction time with an oscillating field becomes comparable to the period, both its transverse and longitudinal momentum distributions may be altered as a function of the electron arrival time within the cycle. Such conditions are often fulfilled at terahertz frequencies, facilitating applications in electron acceleration[14], pulse compression[15,16] and spectral reshaping[17,18].

At even higher frequencies in the optical domain, a further regime is encountered when the electron coherence time exceeds the optical period, or equivalently, the kinetic energy distribution of the electron beam is narrower than one photon energy. For sufficiently strong interaction, inelastic electron–light scattering produces sidebands on the electron energy spectrum[19,20]. Corresponding experiments are typically carried out in an ultrafast transmission electron microscope (UTEM), allowing for femtosecond optical excitations to be probed with a pulsed beam of electrons[19,21–26]. The underlying scattering process was shown to be quantum-coherent in nature[27], which facilitates Ramsey-type sequential interactions[28] yielding holographic interference in imaging[29], as well as the longitudinal and transverse phase modulation of electron beams[30–32], and the generation of attosecond electron pulse trains[33–35]. In the form of Photon-Induced Near-field Electron Microscopy (PINEM)[36], spatial maps of the total transition probability[20,22] or single sideband orders[37–39] yield a direct measure of the optical near-field intensity.

However, the spatially varying phase information associated with the optical field, for example from interfering optical modes, is generally lost in these techniques. In electron energy-loss spectroscopy (EELS), recent progress was made toward accessing phase gradients and nonlocal field correlations during the measurement of

[1]Department of Ultrafast Dynamics, Max Planck Institute for Multidisciplinary Sciences, 37077 Göttingen, Germany. [2]4th Physical Institute – Solids and Nanostructures, University of Göttingen, 37077 Göttingen, Germany. [3]Institute of Numerical and Applied Mathematics, University of Göttingen, 37083 Göttingen, Germany. [4]ICFO-Institut de Ciencies Fotoniques, The Barcelona Institute of Science and Technology, 08860 Castelldefels (Barcelona), Spain. [5]ICREA-Institució Catalana de Recerca i Estudis Avançats, 08010 Barcelona, Spain. ✉e-mail: claus.ropers@mpinat.mpg.de

spontaneous inelastic scattering. Specifically, transverse deflections of a focused probe filtered with EELS in reciprocal space provided a map of the self-induced electron field[40]. Moreover, EELS with tailored and phase-shaped electron beams was shown to be sensitive to two-point field correlations and the symmetries of modes excited by swift electrons[41,42]. In contrast, stimulated interactions used in PINEM, are more advantageous because the illuminating field provides an external phase reference, with a longitudinal and spatial variation imprinted on the electron beam.

In this study, we harness the preservation of coherence in stimulated interactions and introduce Lorentz-PINEM to image both the intensity and phase of optical near fields. In particular, we demonstrate defocus phase contrast in the imaging of standing nanoplasmonic modes on a tapered gold structure. Exploiting the conjugate symmetric relationship between electron energy-gain and -loss sideband orders, we iteratively reconstruct the optically induced electron phase profile.

## Results

### The principle of Lorentz imaging of optical fields

As schematically displayed in Fig. 1, the principle of Lorentz-PINEM combines energy-filtering of sidebands with Fresnel-mode phase-contrast microscopy. For illustration, we discuss a scenario for imaging the fields at a spherical dipolar nanoparticle. Under oblique incidence, p-polarized illumination at frequency $\omega$ induces in- and out-of-plane dipole fields $\mathbf{E}$ around the nanosphere given by $\mathbf{E}(x,y,z,t) = \mathrm{Re}\{\mathbf{E}(x,y,z)e^{-i\omega z/v_e}\}$. The coherent scattering process of stimulated inelastic electron–light scattering is described by the complex coupling coefficient

$$g(x,y) = \frac{e}{2\hbar\omega} \int_{-\infty}^{\infty} E_z(x,y,z)e^{-i\omega z/v_e} \, dz, \qquad (1)$$

representing the electric field component along the electron trajectory at the spatial frequency $\Delta k = \frac{\omega}{v_e}$, where $v_e$ is the electron velocity[19,20]. The interaction generates discrete harmonic sidebands, denoted by the order $N$, in the electron energy spectrum, with amplitudes[20]

$$\Psi_N = J_N(2|g|)e^{iN\arg\{-g\}}. \qquad (2)$$

Notably, gain and loss sidebands of equal order are conjugate symmetric to each other (i.e., $\Psi_{-N} = \overline{\Psi_N}$).

For a dipolar field distribution, an analytical expression for the complex coupling coefficient $g(x,y)$ is given in terms of modified Bessel functions[37,43]. Quantitative spatial maps of the near-field strength are obtained by energy filtering the gain or loss side of the spectrum, recording electron density maps $I(x,y) = \sum_{N \in \mathcal{N}_{\text{filter}}} |\Psi_N|^2$, shown for the nanosphere and $\mathcal{N}_{\text{filter}} = \{1\}$ in Fig. 1b. It is evident that, in the absence of aberrations, the image intensity in focus is independent of the phase of the coupling coefficient $g$. However, under Fresnel-mode (i.e., defocus) Lorentz imaging conditions, phase sensitivity is gained from local phase gradients leading to deflections. The intensity at a defocus distance $\Delta z$ becomes

$$I_{\Delta z}(x,y) = \sum_{N \in \mathcal{N}_{\text{filter}}} \left| \mathcal{D}(\Psi_N) \right|^2, \qquad (3)$$

resulting from the action of the the Fresnel propagator $\mathcal{D}(\Psi)$ on each of the complex sideband amplitudes $\Psi_N(x,y)$. The Fresnel propagator can be expressed in terms of the physical defocus $\Delta z = 960\,\mu m$, or equivalently, using a dimensionless Fresnel number $F$ (see Methods section). In a spatial representation, the propagator is then expressed as convolution in the form $\mathcal{D}(\Psi) := \chi_F * \Psi$, with $\chi_F(x,y) := F/(ia^2) \exp(i\pi F a^{-2}(x^2 + y^2))$. This representation also

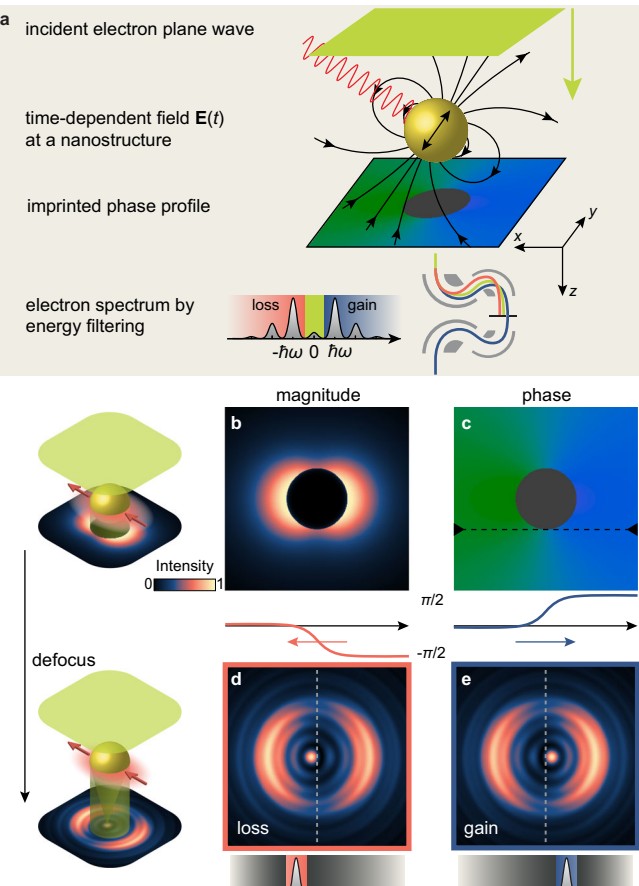

**Fig. 1 | Principle of Lorentz imaging of optical fields. a** A plane-wave electron beam probes the near field of a nanostructure, here a nanosphere with a dipolar field induced by external illumination. Stimulated inelastic scattering populates sidebands in the electron spectrum and imprints a spatial phase profile. **b, c** Simulated magnitude of the first electron sideband (b) after energy-filtering and associated phase profile (c) with lineouts shown below (green = 0 and blue = $\pi/2$; see also phase lineouts of gain and loss sidebands below). **d, e** Simulated Lorentz images under overfocus conditions for loss ($-\hbar\omega$) and gain ($\hbar\omega$) filtered sidebands, exhibiting sensitivity to the phase profile.

allows for a small imaginary part of the Fresnel number, which accounts for a finite transverse coherence length in the experiment and leads to a damping in the propagation of very high spatial frequencies (see Methods section for details).

In Fig. 1d, e, simulated defocused images under overfocus ($\Delta z > 0$) imaging conditions are shown. Fresnel diffraction of the nanosphere produces a bright intensity peak near its centre, the well-known Arago or Poisson spot in the shadow of a circular opaque screen[44]. Additionally, however, the precise location of this spot depends on the local phase gradients in the PINEM field $g(x,y)$, imprinted onto the spatial wave function of the first sideband. Deflected from the geometrical centre of the nanosphere, the peaked feature as well as other more subtle asymmetries exhibit opposite displacements for loss- and gain-filtered images (Fig. 1d,e). As illustrated in this example, phase contrast in Lorentz-PINEM is governed by coupling-induced amplitude contrast and local phase gradients, jointly affecting the Fresnel diffraction within each coherent sideband wave function.

### Imaging polarization-dependent near-field strengths

In the following, we experimentally investigate Lorentz-PINEM contrast and phase retrieval. As a prototypical nanostructure, we study a conical gold nanotip excited by ultrashort near-infrared pulses. Supporting

bound surface-plasmon polariton (SPP) modes and facilitating nanofocusing[45–48], these structures are used in various nano-optics applications, serving as sources for localized ultrafast electron emission[49,50] and as probes in apertureless near-field optical microscopy[51–53]. The measurements are carried out in the Göttingen UTEM featuring a laser-triggered field emitter[54]. Ultrashort near-infrared laser pulses (1.8 ps duration, 800 nm centre wavelength) excite the nanotip sample for variable polarization, while temporally coincident femtosecond electron pulses probe the resulting inelastic scattering using electron imaging and spectroscopy (cf. sketch in Fig. 2a).

For any given structure, the extraction of field-induced phase profiles first requires a precise measurement of the position-dependent scattering probability. To quantitatively characterize this amplitude contrast, we first map the coupling coefficient by recording spectra in scanning TEM (STEM) mode, obtaining $|g(x, y)|$ by a fit to the spectrum at each raster-scanned point (schematic in Fig. 2a). The experimental results are shown in Fig. 2b–e for two linear and two circular illuminating polarizations as an external control parameter. For each polarization, we find a standing wave pattern with nodes and local minima along the tip shaft, evidencing a superposition of bound plasmon modes excited at the apex and the tip-supporting shaft, as well as the scattered field. The symmetry of the distributions, the effective node spacing and position, and the strength of modulation are externally controlled by the incident polarization, and are arising from different coupling efficiencies of the far-field onto the tip. Specifically, the maximum coupling strength at the apex strongly depends on polarization and can be moved from the upper to the lower shaft boundary (Fig. 2b, c). These measurements strongly suggest a polarization-dependent excitation of different azimuthal SPP modes. For example, the asymmetry between top and bottom implies contributions from SPP modes circulating the tip shaft in different directions $m = \pm 1$, where the electron beam typically most strongly couples to the co-propagating fields[39,55,56].

## Lorentz contrast of plasmonic fields at a gold nanotip

With the completed measurement of the magnitude $|g|$, we now turn to obtain the phase-contrast of the optical field. In principle, several measurement options exist, involving different trade-offs in resolution, contrast and signal-to-noise ratio. For example, phase gradients are expected to produce small yet measurable shifts of the diffraction disk associated with the focused probe in energy-filtered STEM, as recently demonstrated by Krehl et al. for spontaneous plasmon excitations[40]. For the coherent population of multiple sidebands by stimulated interaction, as studied here, we found that energy-filtered full-field imaging at a well-selected defocus provides the best practical solution.

We thus switch the electron microscope operation to TEM mode, while keeping the optical excitation of the nanotip unchanged. The near-field interaction with the incident electron plane wave results in a superposition of sideband wave functions, each with an imprinted spatial phase profile (see Eq. (1)), as illustrated in Fig. 3a. Figure 3b shows an energy-filtered ($N > 0$) in-focus image for close to p-polarized optical illumination identical to Fig. 2e. Proving the consistency between the different measurements, apart from a somewhat higher noise level, we find that the image intensity closely corresponds to that predicted from the $g$-map measured by STEM-PINEM (cf. Fig. 2f.)

For comparison with the experiment, we carry out boundary-element method (BEM) simulations (see Methods) for an idealized conical gold nanotip illuminated with parameters (frequency, polarization, angle of incidence) taken from the experiment (Fig. 3c). We find very good qualitative agreement in the modulated pattern arising from SPP excitation and scattering despite the idealised simulation geometry.

Representing a key observation of the present work, Fig. 3d–g displays experimentally measured and simulated energy-filtered images for both loss (d, e) and gain (f, g) filtering under defocus imaging conditions, in a side-by-side comparison. The Lorentz-mode phase-contrast PINEM images exhibit bright spots of enhanced intensity near

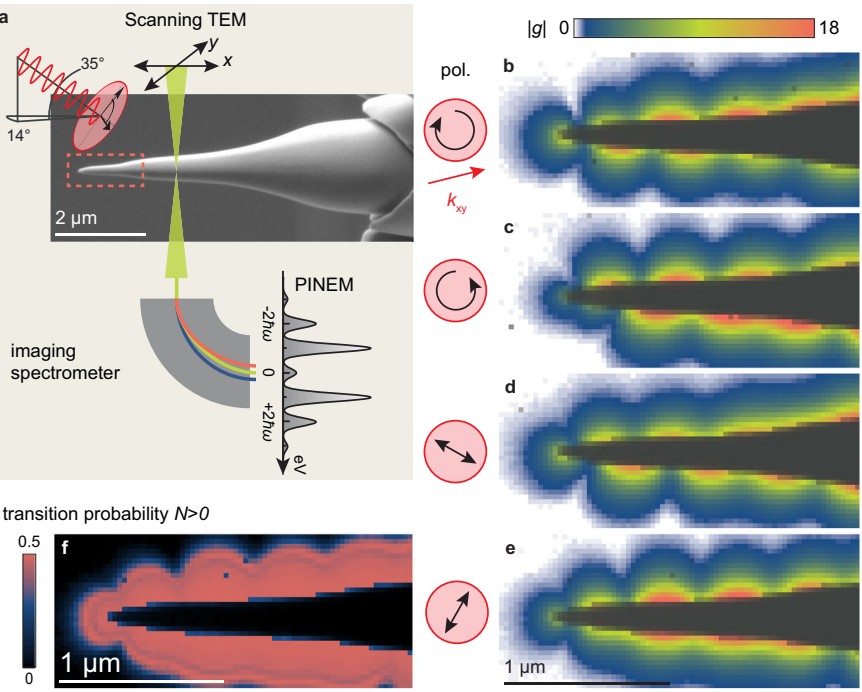

**Fig. 2 | Polarization-dependent near-field strength. a** Scanning TEM imaging of inelastic scattering at a laser-illuminated gold nanotip, recording an electron spectrum at every scanned position. An SEM image shows the nanostructure with the measurement area indicated as a dashed rectangle. **b–e** Maps of the magnitude of the near-field coupling coefficient $|g|$ for different polarizations, extracted from the electron spectrum at each position. The shadow of the tip is overlaid in dark grey. Arrows indicate the polarization of the incident beam and the projected incident light wave vector (red). **f** Total electron scattering probability into gain orders ($N > 0$), obtained from the map shown in (**e**).

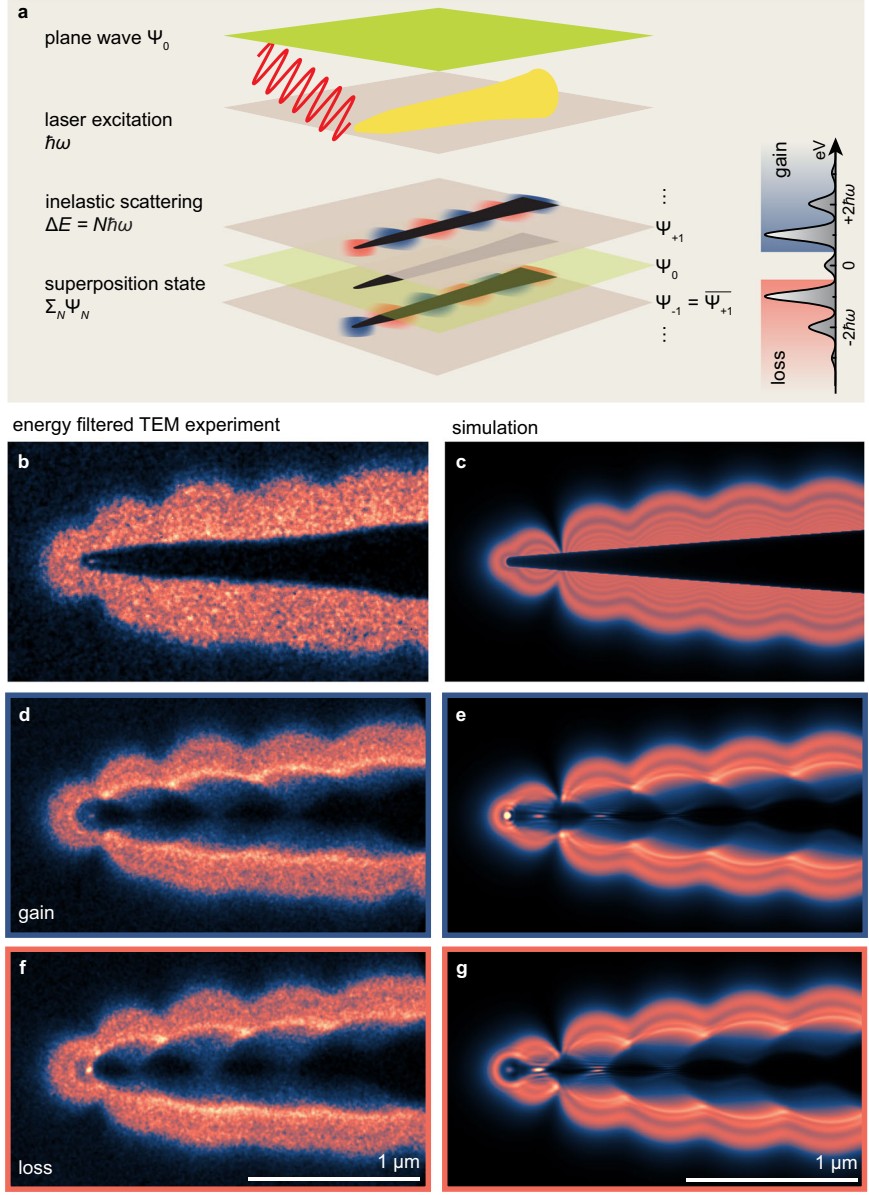

**Fig. 3 | Optical near-field Lorentz contrast at a gold nanotip. a** Schematic of Lorentz-PINEM at the gold nanotip. Inelastic scattering creates a superposition of gain- and loss-scattered sideband wave functions with order-dependent optical phase profiles imprinted. **b** Gain-filtered in-focus TEM image. **c**, Corresponding boundary-element method simulation. **d–g** Measured (**d**, **f**) and simulated (**e**, **g**) defocus Lorentz images of gain (blue frame) and loss (red frame) filtered sidebands, illustrating distinct phase-contrast features at the tip apex and shaft.

the tip apex, as well as caustics along the shaft in both experiment and simulation (Fig. 3d, e). Notably, the images for gain- and loss-filtered electrons significantly differ in the caustics bent towards opposite directions (see also Supplementary Movie 1), and a far weaker bright spot at the apex in gain. These features, qualitatively reproduced by the BEM simulations, are a direct consequence of distinct gain vs. loss phase contrast, as expected from the reversed phases imprinted by the near field.

**Iterative retrieval of an optically induced electron phase profile**

In conventional Lorentz microscopy of magnetic samples, phase-contrast images from a series of known defoci can be used to reconstruct the phase and correspondingly the sample magnetization[57–59]. In our approach, the combination of a |g| map and two correlated defocus micrographs at a single defocus, originating from reversed phases, contains sufficient information for reliable phase retrieval. In the following, we use these data to implement an iterative reconstruction

algorithm of the near-field phase distribution, schematically depicted in the flow chart of Fig. 4a. Specifically, we compare Fresnel-propagated intensities with the experimental data, using an iterative regularized reconstruction approach for ill-posed problems introduced in Ref. 60 which is known to be locally convergent under appropriate assumptions. A similar method has successfully been applied to phase retrieval problems[61] (details of the algorithm are given in Methods).

The algorithm retrieves the complex amplitude of $g(x, y)$ in the focus plane based on the measured defocus gain and loss intensities and the in-focus magnitude $|g(x, y)|$. The reconstruction result is displayed in Fig. 4b, with a lineout of the phase $\arg\{g(x, y)\}$ plotted in Fig. 4c. The phase exhibits a modulated gradient along the shaft, alternating between phase jumps near the nodes of $|g|$ and more extended plateaus at the antinodes. This agrees well with the phase profile of the simulated phase (Supplementary Fig. S5) that is plotted for comparison. In contrast to the behaviour along the shaft, only weak phase changes are found in the radial direction, as expected

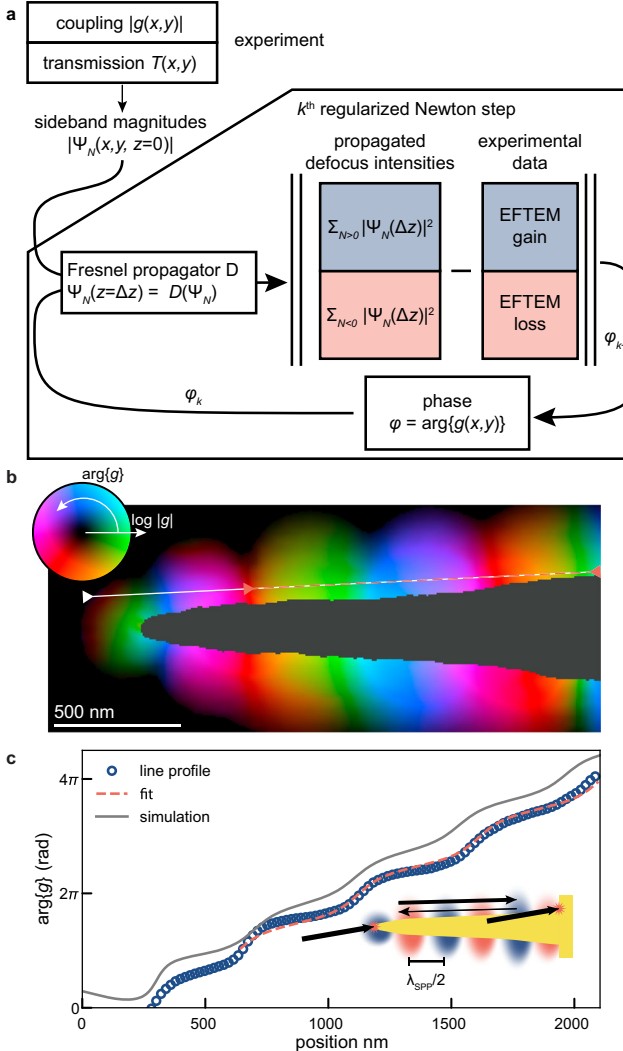

**Fig. 4 | Retrieval of optically induced electron phase profile. a** The regularized reconstruction takes the measured coupling magnitude $|g|$ and the transmission $T$ of the nanostructure as an input, and retrieves the phase $\arg\{g\}$ by comparison of the resulting propagated intensity images with the experimental data, in a regularized iteration. **b** Resulting complex amplitude of the coupling coefficient $g$. **c** Lineout (along the dashed line in (**b**) of the phase, together with fit to a model of counter-propagating waves of unequal amplitude. The sketch indicates the main interfering components of the standing-wave pattern (see text). The lineout of the phase from the BEM simulation (Supplementary Fig. S5) is shown for comparison.

from the exponential decay of the SPP modes. These observations suggest a model of two counter-propagating SPP waves with the same $k$-vector but different amplitudes (dashed line in Fig. 4b), yielding relative amplitudes of 1 : 0.315(15) for forward- and backwards-propagating modes. A comparison of the experiments with BEM simulations suggests that the primary modes responsible for the contrast are $m = 0$ (symmetric contribution with respect to the tip axis) and $m = \pm 1$ (asymmetric contributions), featuring a substantial relative shift (along the shaft) in the position of the nodes of their respective coupling coefficients (cf. Supplementary Fig. S4).

## Discussion

In conclusion, by introducing Lorentz-mode PINEM, we demonstrated the imaging of a spatial phase profile imprinted onto an electron beam by coherent inelastic interaction at a nanostructure. Direct access to two conjugate electron wave functions at a single defocus and identical microscope settings represents a unique and intriguing

characteristic of inelastic phase-contrast imaging. Future implementations may apply the concept to varied electron energies for a full three-dimensional reconstruction of the field. It should be noted that the performance of Lorentz microscopy, and equally our variation thereof, generally involves a compromise between resolution and contrast. Larger defoci yield higher contrast at the expense of lower spatial resolution (cf. Supplementary Fig. S6). A key factor determining the achievable contrast at a given defocus is the coherence of the probing electron beam. In the current implementation, the use of a laser-triggered field emitter in UTEM[54] proves to be an instrumental advantage in resolving the comparatively small phase gradients induced by optical fields. Our approach is broadly applicable and can be used to image arbitrary near fields of current interest such as those in chiral structures[37] and in excitations exhibiting topological character, including optical skyrmions[62–64] and merons[65,66], as well as topologically protected modes[67], bound states in the continuum[68] (to which free electrons can naturally couple), and the $2\pi$ scattering phase shift range required to realize full-range compact metasurfaces[69].

## Methods
### Specimen preparation
The gold nanotip is prepared from an annealed gold wire using focused ion beam milling. The tip has a shaft opening angle of 5.7° and an apex radius of 22.8 nm.

### Ultrafast transmission electron microscopy
The experiments are conducted at the Göttingen UTEM, equipped with a laser-triggered Schottky field emission electron gun delivering 500 fs electron pulses of high spatial coherence[54]. The UTEM allows for the study of ultrafast processes with nanoscale resolution, offering a wide range of imaging techniques and contrast mechanisms[54]. A non-colinear optical parametric amplifier pumped by a regenerative laser amplifier delivers 200 femtosecond laser pulses at 609 kHz repetition rate, 800 nm central wavelength and 131/cm bandwidth which are dispersively stretched to 1.8 ps pulse duration. The nanotip's apex is illuminated with the stretched pulses under an angle of 55° and a focal spot size of 150 µm resulting in a peak intensity of 0.96 GW/cm², as shown in Fig. 2a. A quarter-wave plate and a half-wave plate control the incident polarization which strongly influences the near field at the nanotip as shown in Fig. 2b–e and Supplementary Fig. S2. The incident polarization is measured outside the TEM and then calculated at the specimen using Jones matrices of the incoupling optical components.

We use a post-column imaging energy filter CEFID from CEOS GmbH (technical details can be found in Ref. 70). The CEFID filter is equipped with a single-electron-sensitive hybrid pixel detector (ASI Cheetah with four stitched TimePix3 chips).

### Mapping of near-field strength
The spatial and temporal overlap of the electron and laser pulses is reached by aligning the laser focus to the TEM field of view with a focusing lens on a translation stage, adjusting the timing of the electron and laser pulses with a delay stage. In scanning TEM mode, we record electron energy spectra at every scanned position, with the energy filter set to spectral mode. To extract the coupling constant $|g|$ at every scan position, we fit each measured spectral distribution to the sideband populations $|\Psi_N|^2$, according to Eq. (2), and including an overall prefactor for the transmission and the broadening of the sidebands (i.e., by convolution with the incident kinetic energy spectrum). While such a fit already yields a reliable determination of the spatially varying coupling constant, the spectra are best described by including some variation $\Delta|g|$ of the coupling constant around the respective mean value of $|g|$ at each point, implemented as averaging spectra with a Gaussian distribution of coupling constants. The observed relative uncertainty (standard

deviation over mean) is below 20% for the majority of the spectra measured.

## Energy-filtered Lorentz-PINEM

Lorentz-PINEM (Fig. 1) is implemented in LowMAG TEM mode (indicated magnification ×1000) with a low convergence plane wave illumination. A defocus series of the sample without inelastic electron–light scattering (at a timing delay of -10 ps) is shown in Supplementary Fig. S1. In our experiment, we record images at in- and defocus imaging conditions, at an indicated defocus distance of 192 μm and 960 μm, respectively.

The zero-loss peak spectral width of 0.6 eV yields spectra with clearly identified spectral sidebands separated by an energy of 1.55 eV. The high isochromaticity of the used filter (<30 meV r.m.s. electron energy spread over the 5 mm field of view of the recorded images) allows for homogeneously energy-filtered images. For improved signal-to-noise ratio, in Fig. 3b, d, and f, using motorized slits in the dispersion plane of the filter, we record images integrating over the entire gain and loss regions, respectively, and excluding electrons in the zero-loss peak within a spectral interval of [−0.78, 0.78] eV.

## Image processing

In post-processing, the hot pixels of the camera and the stitching line of the four quadrants are replaced with the average intensities of neighbouring pixels. A drift resulting from the total acquisition time of 40 min per image is compensated by rigid-body image registration, and the changing magnification due to the defocusing is balanced by scaling the images. The scaling factor is determined with the defocus series shown in Supplementary Fig. S1 where we average the intensity around the tip for each image. By assuming that the illuminating electron intensity stays constant we can calculate the scaling factor for each defocus.

To combine the measurements with STEM and LowMag TEM mode, we interpolate the STEM data to the pixel size of the energy-filtered TEM (EFTEM) images. We reference a calculated EFTEM image from the $g$-map to the experimental infocus EFTEM data where we also compensate for slightly rectangular scan pixels (1%)

## Simulation of PINEM coefficient maps

The PINEM coupling coefficient under illumination with quasi-monochromatic light of frequency $\omega$ is given by the integral $g(x,y) = (e/2\hbar\omega) \int dz\, E_z(x,y,z)\, e^{-i\omega z/v_e}$ for each lateral position $(x, y)$ of the electron beam (directed along $z$), where the optical electric-field amplitude is defined such that it yields a time dependence $E_z(x,y,z,t) = \mathrm{Re}\{E_z(x,y,z)e^{-i\omega t}\}$. We adopt a more computationally efficient procedure to calculate $g(x,y)$ by relating it, in virtue of reciprocity, to the cathodoluminescence (CL) far-field $\mathbf{f}(-\hat{\mathbf{k}})\, e^{ikr}/r$ produced by an electron moving with opposite velocity $-\mathbf{v}_e$ along a direction $-\hat{\mathbf{k}}$ (the opposite of the PINEM incident light wave vector direction). Namely[71], $g(x,y) = (ic^2/4\hbar\omega^2)\, \mathbf{f}(\hat{\mathbf{k}}) \cdot \mathbf{E}_0$, where $\mathbf{E}_0$ is the laser-field amplitude. We use BEM to obtain the CL amplitude, modelling the tip as an axially symmetric object to reduce the computation to a one-dimensional self-consistent boundary problem for each azimuthal number $m$ (Ref. 72), and thus cope with the large size of the structure, which we parametrize as a conical tip (25.8 nm apex radius of curvature, 4.35° half-cone angle, 5.255 μm length) supported on a disk (670 nm radius, 200 nm thickness) as shown in Supplementary Fig. S7.

## Iterative phase reconstruction

The task to recover the phase of the coupling coefficient $\arg\{g\}$ in the exterior of the nanotip from the observed intensities $I$ is an ill-posed inverse problem. To fully explain $I$ we have to treat the full complex $g$ in the interior of the nanotip as additional unknowns. More precisely, we chose a function $f = \ln g = \ln|g| + i \arg(g)$ in the interior and $f = i \arg(g)$ in the exterior of the nanotip as unknown of the inverse problem. The

corresponding forward operator $\mathcal{G}$ mapping $f$ onto the data $I$ is built up from the (highly nonlinear) relation in Eq. (2) between $g$ and the amplitudes $\Psi_N$ scaled by the electron transmission, and the incoherent sum in Eq. (3) involving the Fresnel propagator. The image formation is demonstrated in Supplementary Fig. S3. The Fresnel propagator is implemented using a fast Fourier transform (FFT) approximation of the Fourier transform operator $\mathcal{F}$ which allows the calculation of the convolution numerically as multiplication

$$\mathcal{D}(\Psi) := \mathcal{F}^{-1}\left(\exp\left(-\frac{i\pi a^2 k^2}{F}\right)\mathcal{F}(\Psi)(k)\right).$$

The dimensionless Fresnel number is given by

$$F = a^2\left(\Delta z \lambda_e - i\theta_c \Delta z^2 / \ln 2\right)^{-1} \approx 17 + 0.23i$$

using a characteristic sample dimension $a = 200$ nm, defocus $\Delta z = 960$ μm, electron wavelength $\lambda_e = 2.5$ pm and beam divergence semi-angle of the electron source $\theta_c = 5$ μrad (Ref. 73).

The inverse problem $\mathcal{G}(f) = I$ is solved by the Newton-CG method[60], a standard approach in this field. Here the Newton equations $D\mathcal{G}[f_k]\Delta f_k = I - \mathcal{G}(f_k)$ for updates $\Delta f_k = f_{k+1} - f_k$ are solved by the conjugate gradient (CG) method applied to the normal equation $D\mathcal{G}[f_k]^* D\mathcal{G}[f_k]\Delta f_k = D\mathcal{G}[f_k]^*(I - \mathcal{G}(f_k))$. We also obtained very similar results by the iteratively regularized Gauss-Newton method[61,74] where regularization of the Newton equations is achieved by Tikhonov regularization rather than early stopping of the inner CG iteration. In comparison, Newton-CG has the advantage that tuning of an (initial) Tikhonov regularization parameter is not necessary due to scaling invariance of the Newton and the CG methods. The unknown $\ln g$ is penalized by a Sobolev norm of order 2 to enforce smoothness. The data fidelity norm is chosen as a quadratic approximation to the negative Poisson log-likelihood function. Following Morozov's discrepancy priniciple[74], before the Newton step starts to fit the noise, we take the previous step as the result of the reconstruction algorithm.

Computational efficiency has been significantly improved by using a simplified Newton method. We started with the approximate Newton equations $D\mathcal{G}^{(4)}[f_k]\Delta f_k = I - \mathcal{G}(f_k)$ for the Jacobian of a simplified forward operator $\mathcal{G}^{(4)}$ corresponding the filters $\mathcal{N}_{\text{filter}} = \{1,2,3,4\}, \{-1, -2, -3, -4\}$ in (3) as long as these simplified Newton steps reduced the difference between propagated defocus intensities and experimental data by at least a factor of 0.98. When this criterion was violated, we gradually improved the approximation to $\mathcal{G}$ by doubling the number of sidebands and using operators $\mathcal{G}^{(8)}, \mathcal{G}^{(16)}$ and $\mathcal{G}^{(30)} = \mathcal{G}$.

The reconstruction shown in Supplementary Fig. S5 required 11 Newton-CG iterations until termination by the discrepancy principle. It shows that it is possible to reliably reconstruct the phase profile $\arg\{g\}$ in a connected region from out-of-focus intensities averaged over gain and over loss sidebands up to an additive constant. For two or more spatially disconnected fields with disjoint compact supports, there is a further undetermined or at least not stably determined relative phase for each region. For such field configurations, one would likely record an additional measurement with large defocus and low resolution to uniquely determine these relative phases.

## Data availability

The data shown in the manuscript and used for the reconstruction are available on Edmond - the Open Research Data Repository of the Max Planck Society (https://doi.org/10.17617/3.AY9WSD) (Ref. 75).

## Code availability

The reconstruction was carried out using the code available on Edmond - the Open Research Data Repository of the Max Planck Society (https://doi.org/10.17617/3.AY9WSD) (Ref. 75).

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

## Acknowledgements

The authors acknowledge assistance from the Göttingen UTEM team, especially Marcel Möller, Till Domröse, and Thomas Danz. This work was funded by the Deutsche Forschungsgemeinschaft (DFG, German Research Foundation) in the Collaborative Research Centres "Atomic scale control of energy conversion" (217133147/SFB 1073, project A05) and "Mathematics of Experiment" (432680300/SFB 1456, project C01) and via resources from the Gottfried Wilhelm Leibniz Prize (RO 3936/4-1), and the European Union's Horizon 2020 research and innovation programme under grant agreement No. 101017720 (FET-Proactive EBEAM). F.J.G.d.A. acknowledges support from European Research Council (AdG 789104-eNANO) and Spanish MCINN (PID2020-112625GB-I00 and CEX2019- 000910-S).

## Author contributions

J.H.G. conducted the experiments with support from H.L.-M., M.S. and A.F. J.H.G. analysed the data. F.J.G.d.A. performed the finite-element simulations. T.H. wrote the reconstruction algorithm that J.H.G. applied to the experimental data. S.Y. carried out analytical computations. C.R. conceived and directed the study. C.R. and J.H.G. wrote the manuscript, and all authors discussed the results and their interpretation.

## Funding

## Competing interests

The authors declare no competing interests.
