## [Peer Review File · Nature Communications]

REVIEWER COMMENTS

Reviewer #1 (Remarks to the Author):

The manuscript presents a method for real-space mapping of electron phase contrast induced by local optical fields. It is a remarkable demonstration of the way in which light and fast electrons interact. A particularly delightful observation is the observed shift in the Arago spot from defocused electrons that have either gained or lost energy from the optical field.

These are certainly not easy experiments, given the complicated PINEM set-up and the required long acquisition times. The authors are therefore congratulated for obtaining the beautiful results, and I have no objection in seeing this published, after incorporating the minor revisions suggested below.

p.2, Caption of Figure 1: Please indicate what the colours represent (e.g blue = zero, green = 2π) in the phase image of panel 1c.

p.2, equation (2): define N.

p.3, Figure 2: It is not clear how the near-field coupling coefficients are mapped. It is mentioned that they are “extracted from the spectrum at each position” and (in the main text) “by a fit to the spectrum at each raster-scanned point”, or (in the Methods section) “we extract, by fitting, the coupling constant”. However, this does not give enough information for the reader to reproduce the data processing. A clear description of the procedure should be included.

p.2 and 3: In figure 1a, the schematic indicates that the first sideband is filtered and used for mapping. In Figure 3 however, there is no indication of the filtering in the figure. In the caption of Figure 3, it is mentioned that we see either “gain-filtered sidebands” or “loss-filtered sidebands”. Does this mean that the zero-loss peak was simply cut-off at a certain energy and the whole gain-range or loss-range was used for figures 3d and 3f? The Methods-section also does not clarify whether an energy-selecting slit was used for the filtered maps. Please clarify. If indeed no slit was used, but only the zero-loss peak was cut-off, the cutoff-energy should be mentioned.

p.4, Figure 4. This is a nice result of course, but more information should be given about the precision of the phase map near the plasmon resonator. After all, the defocused measurements blur the maps, and by the nature of the measured Lorentz gain and loss maps, the outline of the gold nanotip is not a straight line anymore. Instead, the outline of the gold nanotip has a wave-like appearance, different between the loss map and the gain map, as can be seen in Figures 3d and 3f (and in their simulations). It appears then that along the blurry outline, there are regions where the phase can be determined more reliably than in other regions. It seems that this effect is hidden underneath the grey mask in Figure 4b, but it will be important to include a brief discussion on this in the manuscript text. After all, this effect will determine the spatial resolution and spatial accuracy of the phase map.

Reviewer #2 (Remarks to the Author):

The manuscript by Gaida and co-workers reports on a novel electron-beam based technique which allows for the reconstruction of the amplitude and phase of the optical near-field around an illuminated nanostructure. In a clever experimental design, the authors utilize a combination of scanning photon-induced near-field electron microscopy (PINEM) and in-line electron holography (Lorentz microscopy) of energy-filtered electron micrographs for the gain or loss side of the PINEM-broadened electron energy spectrum. The former yields the local amplitude of the optical near-field and the latter relates the phases of the transmitted electron wave function in transversely adjacent regions. I believe this novel electron microscopy method is highly interesting for researchers in the broad field of nano-optics and the manuscript would be suitable for Nature Communications. After properly addressing the questions and comments listed below, I would therefore recommend publication of the manuscript with minor modifications.

Questions/comments:

- (1) For corroborating the retrieved phase maps based on the experimental data, it would be beneficial to also show the phase maps from the boundary-element method (BEM) calculations.
- (2) In the discussion related to Fig. 4c, the oscillatory coupling strength along the tip, is attributed to a standing wave pattern of counter-propagating surface plasmon polariton (SPP) waves originating from the apex of the tip and from some additional discontinuity further down the shaft of the tip. In this picture, the location of the field maxima should be connected to the distance of this discontinuity from the apex. Is this observed experimentally? How was this scenario implemented in the BEM simulation? What is the relative PINEM contribution from incident and reflected propagating waves (i.e. of non-interface-bound wave-field components)?
- (3) What determines the orientation of the line in Fig. 4b along which the experimental phase is extracted (Fig. 4c), since the line seems to not run parallelly to the tip surface? How does the BEM calculated phase compare to this line profile?
- (4) In the abstract, near-field imaging at high spatial resolution is claimed. What is the currently achievable spatial resolution with this technique given the loss of resolution due to image defocusing and the rather long 40-min acquisition time per image? Furthermore, since only a certain Fourier component of the field component along the electron-beam direction is imaged, the method is only indirectly giving access to the phase distribution of the near-field. Direct comparison between the relative phases in a near-field and the retrieved PINEM phase seems to be only possible if the field profile along the beam trajectory is independent of the lateral position.
- (5) Is there a unique solution to the reconstructed phase of the coupling constant for the present case? If yes, do the authors also expect a unique phase solution if spatially separated optical near-fields are sampled? Since the Fresnel propagator only superposes nearby electron wave-field components, I would expect that there should be no information on the relative phase of non-connected near-fields contained in the Lorentz micrographs.

(6) In the experiments, energy filtering micrographs contain the contribution from several photon-side bands. In particular for near-field regions with larger coupling strength, I would expect that some loss of information occurs, due to averaging over the different phase shifts in the varying photon side bands. It would be beneficial if the authors would comment on this.

Minor comments and questions:

(1) What is the (projected) direction of the incident optical wave vector in Fig.2(b-e)?

(2) What determines the chosen image defocus length?

(3) In the simulation results in Fig. 3(c,e,g), the energy filtered Lorentz micrographs seem to show an approximate mirror symmetry along the gold tip axis. This is surprising since the tip is illuminated under an oblique angle relative to the direction of the electron beam, which should break the mirror symmetry in the micrographs. Was this configuration deliberately chosen experimentally based on the expected symmetry from the simulation? How did the authors determine the optical polarization at the sample?

(4) The detailed comparison between experimental and simulated maps of coupling strengths for different light polarizations (Fig. 6) is only partially convincing. Is the observable difference related to the idealized geometry in the BEM calculation?

Response to Reviewers

We thank all Reviewers for spending time with our work and for the constructive comments and questions.

Reviewer #1 (Remarks to the Author):

Reviewer #1 states:

The manuscript presents a method for real-space mapping of electron phase contrast induced by local optical fields. It is a remarkable demonstration of the way in which light and fast electrons interact. A particularly delightful observation is the observed shift in the Arago spot from defocused electrons that have either gained or lost energy from the optical field.

These are certainly not easy experiments, given the complicated PINEM set-up and the required long acquisition times. The authors are therefore congratulated for obtaining the beautiful results, and I have no objection in seeing this published, after incorporating the minor revisions suggested below.

p.2, Caption of Figure 1: Please indicate what the colours represent (e.g blue = zero, green = 2π) in the phase image of panel 1c.

Our Response:

We thank the Reviewer for assessing our manuscript, and for the very positive comments. As for the missing colour labels, we thank the Reviewer for noticing this omission, and we have revised the caption accordingly.

Reviewer #1 states:

p.2, equation (2): define N .

Our Response:

We added the definition to the updated version.

Reviewer #1 states:

p.3, Figure 2: It is not clear how the near-field coupling coefficients are mapped. It is mentioned that they are “extracted from the spectrum at each position” and (in the main text) “by a fit to the spectrum at each raster-scanned point”, or (in the Methods section) “we extract, by fitting, the coupling constant”. However, this does not give enough information for the reader to reproduce the data processing. A clear description of the procedure should be included.

Our Response:

Thank you for pointing this out. We agree that further explanation is warranted. In the revision, we added a more detailed description of the fit in the methods section.

Reviewer #1 states:

p.2 and 3: In figure 1a, the schematic indicates that the first sideband is filtered and used for mapping. In Figure 3 however, there is no indication of the filtering in the figure. In the caption of Figure 3, it is mentioned that we see either “gain-filtered sidebands” or “loss-filtered sidebands”. Does this mean that the zero-loss peak was simply cut-off at a certain energy and the whole gain-range or loss-range was used for figures 3d and 3f? The Methods-section also does not clarify

whether an energy-selecting slit was used for the filtered maps. Please clarify. If indeed no slit was used, but only the zero-loss peak was cut-off, the cutoff-energy should be mentioned.

Our Response:

The Reviewer is fully correct that the simulation in Fig. 1 displays the result for the first sidebands only. For improved signal-to-noise ratio, in the experiments shown in Fig. 3, we cut off the zero-loss peak and integrate over the entire gain and loss regions, respectively. In the current manuscript, this filtering is indicated by a sketch at the right side of Fig. 3. We now also clarify these points in the methods and mention the excluded interval around the zero-loss peak.

Reviewer #1 states:

p.4, Figure 4. This is a nice result of course, but more information should be given about the precision of the phase map near the plasmon resonator. After all, the defocused measurements blur the maps, and by the nature of the measured Lorentz gain and loss maps, the outline of the gold nanotip is not a straight line anymore. Instead, the outline of the gold nanotip has a wave-like appearance, different between the loss map and the gain map, as can be seen in Figures 3d and 3f (and in their simulations). It appears then that along the blurry outline, there are regions where the phase can be determined more reliably than in other regions. It seems that this effect is hidden underneath the grey mask in Figure 4b, but it will be important to include a brief discussion on this in the manuscript text. After all, this effect will determine the spatial resolution and spatial accuracy of the phase map.

Our Response:

We thank the Reviewer for bringing up this point. There are two aspects which should be considered. First, the wavy outline in the measured gain and loss spectra do not by themselves affect negatively the resolution or accuracy of the reconstruction. They can perhaps be better thought of as a necessary feature of the experimental data that are input for the phase reconstruction. The reconstruction directly yields the phase distribution in the sample plane, i.e., where the image is in focus. The wavy nature of the outline is a result of the phase gradients imprinted in the sample plane and propagated to the defocus plane, but the retrieved phase information is displayed in-focus, together with the measured magnitude of the coupling constant. We now clarify this point in the main text.

Second, we acknowledge that further information about resolution will be helpful to the reader. Generally, the imaging resolution depends on the phase-contrast transfer function, which for the experimental parameters, we added in a new extended data Fig. 10 for different defocus values.

We thank the Reviewer again for the insightful comments, which we believe have resulted in an improvement of the manuscript.

Reviewer #2 (Remarks to the Author):

Reviewer #2 states:

The manuscript by Gaida and co-workers reports on a novel electron-beam based technique which allows for the reconstruction of the amplitude and phase of the optical near-field around an illuminated nanostructure. In a clever experimental design, the authors utilize a combination of scanning photon-induced near-field electron microscopy (PINEM) and in-line electron holography (Lorentz microscopy) of energy-filtered electron micrographs for the gain or loss side of the PINEM-broadened electron energy spectrum. The former yields the local amplitude of the optical near-field and the latter relates the phases of the transmitted electron wave function in transversely adjacent regions. I believe this novel electron microscopy method is highly interesting for researchers in the broad field of nano-optics and the manuscript would be suitable for Nature Communications. After properly addressing the questions and comments listed below, I would therefore recommend publication of the manuscript with minor modifications.

Our Response:

We thank the Reviewer for a careful review of our manuscript and the very supportive comments.

Questions/comments:

Reviewer #2 states:

(1) For corroborating the retrieved phase maps based on the experimental data, it would be beneficial to also show the phase maps from the boundary-element method (BEM) calculations.

Our Response:

This is a very good suggestion, and we have followed the Reviewer's advice to add the phase maps of the simulations to the Extended Data (Fig. 9).

Moreover, to corroborate further our findings and the developed algorithm, we used the simulated defocus intensities to successfully reconstruct these simulated complex amplitudes (also shown in Extended Data Fig. 9).

Reviewer #2 states:

(2) In the discussion related to Fig. 4c, the oscillatory coupling strength along the tip, is attributed to a standing wave pattern of counter-propagating surface plasmon polariton (SPP) waves originating from the apex of the tip and from some additional discontinuity further down the shaft of the tip. In this picture, the location of the field maxima should be connected to the distance of this discontinuity from the apex. Is this observed experimentally? How was this scenario implemented in the BEM simulation? What is the relative PINEM contribution from incident and reflected propagating waves (i.e. of non-interface-bound wave-field components)?

Our Response:

We thank the Reviewer for these very relevant questions. The Reviewer is correct that the specific interference conditions will depend on the distance of the additional scatterer outside the field of view. In the experiments, the rugged shape of the tip shaft (Fig. 1a) makes it somewhat difficult to assess the precise geometry. Therefore, in the BEM simulations, we added a generic scatterer in the form of a larger platform truncating the tip shaft. We varied the size of this platform from the apex to approximately match the modulation pattern and depth observed experimentally. We added a description and a display of the simulated BEM geometry to the Extended data (Fig. 11).

As for the different near-field contributions, the PINEM signal is dominated by propagating SPPs on the tip shaft, with the relative contribution of the mode propagating away from the apex to the one propagating towards the apex being about 1 : 0.29. This value is given in the current main text of the manuscript.

Reviewer #2 states:

(3) What determines the orientation of the line in Fig. 4b along which the experimental phase is extracted (Fig. 4c), since the line seems to not run parallelly to the tip surface? How does the BEM calculated phase compare to this line profile?

Our Response:

As the tip shape is not perfectly conical and there are some changes in opening angle, we decided to choose a lineout that is parallel to the shaft throughout a large portion of the field of view. The phase of the BEM simulations agrees well with this profile, as seen in Figure 4 c. We have added corresponding comments to the main text.

Reviewer #2 states:

(4) In the abstract, near-field imaging at high spatial resolution is claimed. What is the currently achievable spatial resolution with this technique given the loss of resolution due to image defocusing and the rather long 40-min acquisition time per image? Furthermore, since only a certain Fourier component of the field component along the electron-beam direction is imaged, the method is only indirectly giving access to the phase distribution of the near-field. Direct comparison between the relative phases in a near-field and the retrieved PINEM phase seems to be only possible if the field profile along the beam trajectory is independent of the lateral position.

Our Response:

In order to give a quantitative representation of the achievable resolution, we added phase-contrast transfer functions for different defocus values to the Extended Data.

As for the sensitivity to a given Fourier component, we agree that this is a certain limitation. The Reviewer is correct that the spatial variation of the out-of-plane field profile may affect the relation between the retrieved component and other quantities describing the field, such as its magnitude and phase in a given plane. In the future, measurements at different electron energies could be used to carry out a complete three-dimensional reconstruction of the field distribution. We noted this possibility in the conclusions.

Concerning the required measurement time, we experienced that the experiment is sufficiently stable on this length scale during a time of one minute. For the longer measurement times, we then used drift correction by image registration before summing frames.

Reviewer #2 states:

(5) Is there a unique solution to the reconstructed phase of the coupling constant for the present case? If yes, do the authors also expect a unique phase solution if spatially separated optical near-fields are sampled? Since the Fresnel propagator only superposes nearby electron wave-field components, I would expect that there should be no information on the relative phase of non-connected near-fields contained in the Lorentz micrographs.

Our Response:

We thank the Reviewer for this interesting question. Indeed, as a form of in-line holography, our scheme relies on the interference of electron trajectories, and we determine the relative phases of these paths. For connected near-fields, the added reconstruction for simulated data with added noise in the new Fig. 9 present numerical evidence that the phase is uniquely determined by our data up to a single global phase, and that the inverse problem is reasonably stable. This makes us confident of the reliability of our approach. Up to now, we have not derived a mathematical proof of these statements in the case of imaging incoherent sums of sidebands. If filters for single sidebands were used, uniqueness and stability would follow from our previous results in Maretzke & Hohage (2017) and Maretzke (2015).

For two or more spatially disconnected fields with disjoint compact supports, there is a further undetermined relative phase for each region. (Strictly speaking, the Fresnel propagator is global, but since coupling of distant points is extremely weak, it is clear that even if the inverse problem should mathematically have a unique solution, the stability of relative phases in well separated regions would be so weak that such uniqueness would be practically irrelevant.) For such field configurations, one would likely record an additional measurement with large defocus and low resolution to uniquely determine these relative phases. We added corresponding statements to the manuscript.

Reviewer #2 states:

(6) In the experiments, energy filtering micrographs contain the contribution from several photon-side bands. In particular for near-field regions with larger coupling strength, I would expect that some loss of information occurs, due to averaging over the different phase shifts in the varying photon side bands. It would be beneficial if the authors would comment on this.

Our Response:

The Reviewer is certainly right that by summing over contributions from different photon-side bands, information may be lost. Certainly, purely Poisson-distributed data for individual side bands contain at least as much information as data with the same total number of electrons averaged over all loss and all gain side bands. However, the phase profiles of different orders are closely related to each other, scaled only by a factor of N . Therefore, phase gradients lead to deflections in the same direction for all orders with the same sign of N (i.e. gain vs. loss orders), as one would expect for classical trajectories.

As a cross-check, we have applied our algorithm used for reconstructions from simulated data in Figure 9 also to simulated data for individual side bands. Whereas for a given pointwise signal-to-noise ratio, the quality of the resulting reconstructions improved, for a fixed total number of total electrons, the quality of the reconstruction results was comparable, both concerning visual inspection and error norms. This may be caused by the fact that for a given total number of electrons, due to the larger number of micrographs for the reconstruction of individual orders, the pointwise signal-to-noise ratio is reduced, and a non-quadratic approximation of the negative Poisson log-likelihood (the Kullback-Leibler divergence) would have been required to take advantage of the full information content of the data. However, such non-quadratic data fidelity terms involving non-negativity constraints require more complicated algorithms and entail considerable additional computational burdens.

We have commented on the reconstruction using individual sidebands in the revised manuscript and plan to further investigate the details of convergence and loss of information by incoherent averaging over orders in future work.

Minor comments and questions

Reviewer #2 states:

(1) What is the (projected) direction of the incident optical wave vector in Fig.2(b-e)?

Our Response:

Thank you for the question. We agree this information is helpful to the reader. We have added a projected incidence vector to Fig. 2.

Reviewer #2 states:

(2) What determines the chosen image defocus length?

Our Response:

This is a very relevant question. As in other in-line holographic experimental approaches, the defocus is a trade-off between contrast and resolution. We now state this more explicitly in the manuscript. We also added phase-contrast transfer functions in Fig. 10.

Reviewer #2 states:

(3) In the simulation results in Fig. 3(c,e,g), the energy filtered Lorentz micrographs seem to show an approximate mirror symmetry along the gold tip axis. This is surprising since the tip is illuminated under an oblique angle relative to the direction of the electron beam, which should break the mirror symmetry in the micrographs. Was this configuration deliberately chosen experimentally based on the expected symmetry from the simulation? How did the authors determine the optical polarization at the sample?

Our Response:

Generally, the tip is illuminated nearly from the front, at a weakly inclined in-plane angle and a larger out-of-plane angle. We believe that the approximate symmetry arises from the fact that a considerable part of the near-field is composed of propagating surface plasmon polaritons generated near the apex, which is a point-like defect.

The incident angles in the simulation were set up to match those in the experiment. The incident polarization was measured outside the TEM and then calculated using Jones matrices of the optical components used for coupling light to the sample region. We added corresponding descriptions in the Methods section.

Reviewer #2 states:

(4) The detailed comparison between experimental and simulated maps of coupling strengths for different light polarizations (Fig. 6) is only partially convincing. Is the observable difference related to the idealized geometry in the BEM calculation?

Our Response:

As a result of the somewhat irregular geometry of the investigated structure in the shaft region, not all features are reproduced by the simulation. We decided to compare our experimental data with an idealized tip geometry to illustrate the main characteristics of the field, but not all features. In the revised version of the manuscript, we added a note stating that some deviations arise from the imperfect structure studied.

We thank the Reviewer again for the insightful comments, and we hope the Reviewer finds the manuscript suitable for publication in its present form.

REVIEWERS' COMMENTS

Reviewer #1 (Remarks to the Author):

I thank the authors for addressing all the points that were raised to the first version of the manuscript.

I have no further questions or requests anymore, and look forward to the published version of this nice work.